# Association between vitamin D level and respiratory distress syndrome: A systematic review and meta-analysis

Yoo Jinie Kim[1], Gina Lim[2], Ran Lee[1,3], Sochung Chung[1,3], Jae Sung Son[1,3], Hye Won Park[1,3]*

1 Department of Pediatrics, Konkuk University Medical Center, Seoul, Republic of Korea, 2 Department of Pediatrics, Ulsan University Hospital, University of Ulsan College of Medicine, Ulsan, Republic of Korea, 3 Konkuk University School of Medicine, Seoul, Republic of Korea

* 20110673@kuh.ac.kr

**Data Availability Statement:** All relevant data are within the paper and its Supporting information files.

## Abstract

### Background

Growing evidence suggests an association between the vitamin D levels and respiratory outcomes of preterm infants. The objective of this systematic review and meta-analysis was to explore whether premature neonates with a vitamin D deficiency have an increased risk of respiratory distress syndrome (RDS).

### Methods

We searched PubMed, EMBASE, and the Cochrane Library up through July 20, 2021. The search terms were 'premature infant', 'vitamin D', and 'respiratory distress syndrome'. We retrieved randomized controlled trials and cohort and case-control studies. For statistical analysis, we employed the random-effects model in Comprehensive Meta-Analysis Software ver. 3.3. We employed the Newcastle-Ottawa Scales for quality assessment of the included studies.

### Results

A total of 121 potentially relevant studies were found, of which 15 (12 cohort studies and 3 case-control studies) met the inclusion criteria; the studies included 2,051 preterm infants. We found significant associations between RDS development in such infants and vitamin D deficiency within 24 h of birth based on various criteria, thus vitamin D levels < 30 ng/mL (OR 3.478; 95% CI 1.817–6.659; $p < 0.001$), < 20 ng/mL (OR 4.549; 95% CI 3.007–6.881; $p < 0.001$), < 15 ng/mL (OR 17.267; 95% CI 1.084–275.112; $p = 0.044$), and < 10 ng/ml (OR 1.732; 95% CI 1.031–2.910; $p = 0.038$), and an even lower level of vitamin D (SMD = −0.656; 95% CI −1.029 to −0.283; $p = 0.001$).

### Conclusion

Although the vitamin D deficiency definitions varied and different methods were used to measure vitamin D levels, vitamin D deficiency or lower levels of vitamin D within 24 h of

**Funding:** All authors received no specific funding for this work.

**Competing interests:** All authors have declared that no competing interests exist.

birth were always associated with RDS development. Monitoring of neonatal vitamin D levels or the maintenance of adequate levels may reduce the risk of RDS.

## Introduction

Vitamin D participates in mineral-ion homeostasis (e.g., calcium and phosphorus) and bone metabolism [1]. Apart from these classical functions, a role for vitamin D in lung development has been demonstrated by several studies (including animal works) [1–12]. Vitamin D regulates cell proliferation/differentiation, apoptosis, and angiogenesis in the lungs [6] and elsewhere in the body [1]. A recent meta-analysis found an association between the vitamin D level within 24 h of birth and the risk of bronchopulmonary dysplasia (BPD) in premature neonates [13].

Vitamin D receptors are expressed mainly in the lung during late pregnancy [2]; vitamin D thus affects lung functional and anatomical development, including cell differentiation and surfactant synthesis/secretion [3, 5, 14]. Animal studies performed by Nguyen *et al*. [7–9] and Marin *et al*. [2, 10] using rat models revealed high-level expression of vitamin D receptor on type II alveolar cells at the end of gestation, when alveolar cell differentiation and surfactant synthesis commence [2, 10]. Rehan *et al*. [3] reported that the active form of vitamin D enhanced surfactant synthesis by type II pulmonary cells. Surfactant phospholipid and protein B synthesis increased in human pulmonary cancer cell lines. Such laboratory-based work raised the question of whether the respiratory distress syndrome (RDS) risk is increased in preterm neonates with vitamin D deficiencies in clinical settings.

Several clinical studies have described low vitamin D levels in preterm infants who suffered from RDS [15, 16]. However, Matejek *et al*. [17] found no association between vitamin D deficiency and RDS. Given such conflicting evidence, we performed a systematic review and meta-analysis to explore whether vitamin D deficiency or the vitamin D level measured within 24 h of birth was associated with an increased risk of RDS.

## Methods

### Search strategy and study selection

This meta-analysis was reported based the Preferred Reporting Items for Systematic Reviews and Meta-Analyses (PRISMA) recommendations [18, 19] (S1 Table). We searched PubMed, EMBASE, and the Cochrane Library using the following search terms: ['premature infant' or preterm or newborn or neonate or 'low birth weight infant' or 'very low birth weight infant' or 'extremely low birth weight infant'] and ['Vitamin D' or '25-hydroxyvitamin D' or 25-hydroxyergocalciferol or ergocalciferol or cholecalciferol or hydroxycholecalciferol or calcifediol or dihydroxycholecalciferol or 25(OH)D or 1,25(OH)2-vitD], and ['respiratory distress syndrome' or RDS or 'hyaline membrane disease' or 'transient tachypnea' or 'pulmonary surfactants']. Additionally, we manually checked all reference lists in an effort to identify additional relevant studies. The last search was performed on July 20, 2021. We did not place any restrictions, including on language.

We initially reviewed the titles and abstracts of the articles, and then reviewed the full-text articles. The reviews were performed independently by two reviewers (HW Park and Y Kim) using criteria that we determined before the review process for inclusion in the meta-analysis. Any disagreement during the process was resolved by the third author (R Lee).

## Inclusion and exclusion criteria

We chose to include studies that satisfied the following criteria. Study design: A randomized controlled trial, or a cohort study (prospective or retrospective), or a case-control study; Patients and interventions (exposures): Newborns for whom vitamin D levels were known; Outcome: RDS. We excluded case reports, case series, editorials, review articles, and letters. Articles with inadequate or irrelevant data for analysis were also excluded after reading the full text.

## Outcomes

The primary outcome was neonatal RDS diagnosed via chest radiographic findings and clinical presentations (e.g., tachypnea, nasal flaring, chest retraction, and cyanosis) within hours of birth.

## Data extraction

We extracted the data from all included studies via full text review. Two authors (Y Kim and HW Park) reviewed and extracted data using a pre-prepared form. The data were the first author, publication year, study period, study design, study location, study population, sample size, method of vitamin D measurement, definition of vitamin D deficiency, diagnostic criteria for RDS, sample size, number of patients diagnosed with RDS or vitamin deficiency, and vitamin D levels or odds ratios of RDS if vitamin D was deficient for RDS, when possible. If there were any discrepancies during the review process with two authors (YJ Kim and HW Park), we discussed these with the third reviewer (R Lee) and we reviewed the study again.

## Study quality assessment

We (Y Kim and HW Park) also separately assessed the quality of included studies using the Newcastle–Ottawa Scale (NOS) [20]. Disparities in the assessment were resolved through discussion with the third author (R Lee). This scale is composed of three domains that explore selection, comparability, and outcome. The three domains contain a total of eight items. We gave one star to an item that met the criteria, except for comparability (two stars). The score range is 0–9, and the total score indicates the methodological quality of the study; $\leq 3$ is low, 4–5 is moderate, and $\geq 6$ is high. The scores of each study for each items of NOS are provided in the S2 Table.

## Data synthesis and statistical analyses

We performed this meta-analysis to calculate a pooled estimate of odds ratio (OR) for the association between vitamin D deficiency and the occurrence of RDS. We performed the analysis separately when studies reported more than one result based on different definitions of vitamin D deficiency; we thus kept the statistical assumption of the independence of an effect.

We used the $I^2$ statistic to evaluate statistical heterogeneity; the value is expressed as the percentage of total variation across studies. If the $I^2$ value is greater than 50%, this indicates the presence of significant heterogeneity across the studies. We performed our analysis conservatively; based on estimation of the between-study variations in effect size, we used a random-effects model, which yields wider CIs than a fixed-effects model [21]. We also conducted sensitivity analyses to evaluate the effect of each study on the robustness of the combined estimates and contribution to the pooled OR. A cumulative analysis was conducted by adding one study at a time, by year of publication, to evaluate temporal trends.

To detect publication bias, the Begg and Mazumdar rank-correlation test and Egger's regression test were used. Publication bias was also evaluated based on the distribution of the effect sizes against the standard errors on a graphically displayed funnel plot. We detected publication bias by inspecting the funnel plot. Asymmetry of the funnel plot or a P-value < 0.05 in the Begg and Mazumdar rank-correlation test or Egger's regression test were taken to indicate the presence of publication bias. The current meta-analysis was conducted using Comprehensive Meta-Analysis software version 3.3 (Biostat Inc., Englewood, NJ, USA).

## Results

### Literature search and study selection

We show the flow of study selection and exclusion in Fig 1. In total, 38 duplicates were removed from the 121 studies retrieved in the initial search, including the manual search. During review of abstracts and titles, 48 studies were removed, leaving 35 studies. After full-text review, 20 studies were excluded for the reasons described in Fig 1, leaving 15 studies for meta-analysis.

### Characteristics of the included studies

The characteristics of the 15 studies [5, 11, 15–17, 22–31] included in this meta-analysis are shown in Table 1. When evaluating the association between vitamin D deficiency and RDS,

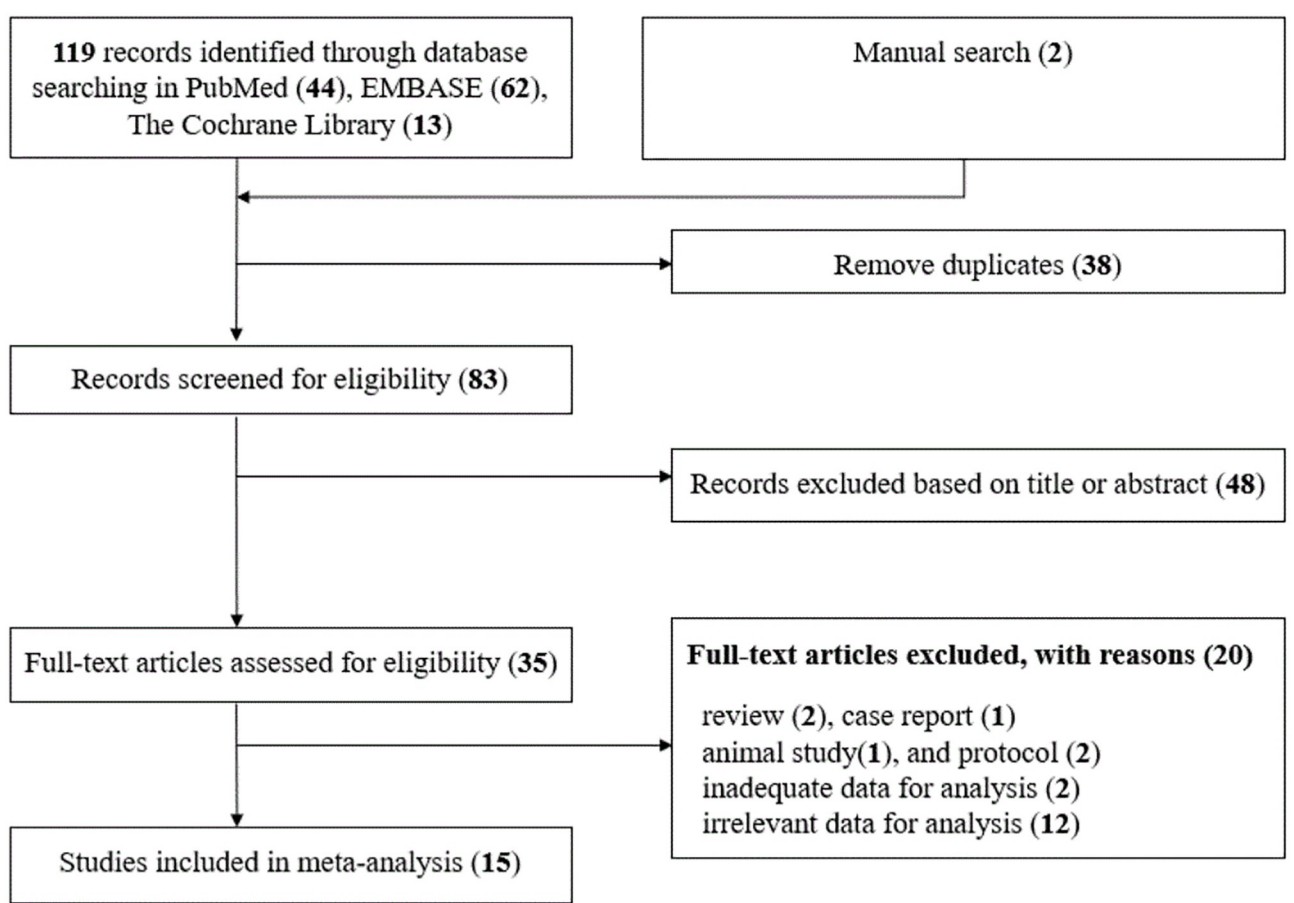

**Fig 1. Flow chart of study selection process.**

**Table 1. Characteristics of studies included in the meta-analysis.**

| Studies | Study design | Nation | Study population | Time of vitamin D measurement | Methods of measurement of vitamin D | Definition of vitamin D deficiency (serum 25(OH)D level) | NOS |
|---|---|---|---|---|---|---|---|
| Ataseven et al. [22] | prospective cohort | Turkey | GA 29–35 weeks | < 24 h after birth | LC-MS | severe: <10 ng/mL moderate: 10–20 ng/mL mild: 20–30 ng/mL | 8 |
| Fettah et al. [23] | prospective cohort | Turkey | GA < 32 weeks | soon after birth (cord) | ELISA | <15 ng/mL | 7 |
| Onwuneme et al. [24] | prospective cohort | Ireland | GA <32 weeks or BW <1,500g | < 24 h after birth | chemiluminescence | <12 ng/mL | 5 |
| Yu et al. [15] | retrospective cohort | China | GA<32 weeks | <24 h after birth | chemiluminescence | <20 ng/ml | 8 |
| Mohamed et al. [25] | prospective case–control | Egypt | GA <34 weeks | < 24 h after birth | ELISA | <20 ng/ml | 5 |
| Yang et al. [26] | retrospective cohort | China | GA <37 weeks | 1st sampling after birth | automatic biochemical analyzer | measured value* | 4 |
| Boskabadi et al. [27] | prospective case-control | Iran | GA < 34 weeks and BW < 2,000g | NA | ELISA | severe: <10 ng/mL moderate: 10–20 ng/mL mild: 20–30 ng/mL | 7† |
| Kazzi et al. [28] | prospective cohort | USA | Preterm infants and BW ≤1,250 g | < 24 h after birth | LC-MS | ≤10 ng/mL | 5 |
| Kim et al. [29] | retrospective cohort | Korea | BW <1,500 g | < 24 h after birth | LC-MS and chemiluminescence | severe: <10 ng/mL deficiency: 10–20 ng/mL insufficiency: 20–30 ng/mL | 8 |
| Treiber et al. [30] | prospective cohort | Slovenia | newborn | soon after birth (cord) | chemiluminescence | severe deficiency: <10ng/mL deficiency: 10–20 ng/ml insufficiency: 20–30 ng/ml | 4 |
| Ardastani et al. [5] | prospective cohort | Iran | GA 28–37 weeks and BW ≥1,000 g | soon after birth (cord) or <24h after birth | ELISA | severe deficiency: <10ng/mL deficiency: 10–20 ng/ml inadequate: 20–30 ng/ml | 7 |
| Matejek et al. [17] | prospective cohort | Czech | BW <1,500 g | soon after birth (cord) | LC-MS | <10 ng/mL | 7 |
| Al-Beltagi et al. [31] | prospective case-control | Egypt | Preterm infants | soon after birth (cord) | ELISA | measured value* | 5† |
| Dogan et al. [16] | prospective cohort | Turkey | GA ≤32 weeks | < 6 h after birth | LC-MS | severe: <5 ng/mL moderate: 5–15 ng/mL mild:15–30 ng/mL | 7 |
| Zhang et al. [11] | prospective cohort | China | GA< 32 week | < 24 h after birth | chemiluminescence | severe deficiency: <10ng/mL deficiency: 10–20 ng/ml insufficiency: 20–30 ng/ml | 7 |

* The measured value of vitamin D was used in the analysis

† The two case-control studies were assessed using the Newcastle-Ottawa Scale for quality assessment of case-control studies; the other studies were assessed using the Newcastle-Ottawa Scale for the assessment of cohort studies

**Abbreviations**: GA, gestational age at birth; BW, birth weight; g, gram; NOS, Newcastle–Ottawa Scale, LC-MS, liquid chromatography-tandem mass spectrometry; ELISA, enzyme linked immunosorbent assay

2,051 infants were included. The mean birth weight of the study population in the 15 included studies was 1,833.2 g (standard error 143.91), and the mean gestational age of the infants at birth was 32.0 weeks (standard error 0.90); one study [26] did not describe gestational age at birth.

The definition of vitamin D deficiency in each study is shown in Table 1. The occurrence rates of RDS are presented by reference to the severity of vitamin D deficiency. Thus, we used the various levels in meta-analysis: a vitamin D level below 30 ng/ml in two studies [5, 27], 20

ng/ml in six studies [11, 15, 25, 27, 29, 30], 15 ng/ml in two studies [16, 23], and 10 ng/ml in nine studies [11, 17, 22, 24, 25, 28–30, 32].

The scores on the Newcastle–Ottawa Scale for quality assessment of all studies are shown in Table 1.

## Pooled meta-analysis results

There were significant associations between vitamin D deficiency and RDS, regardless of the definition of vitamin D deficiency; thus at cut off values of 30 ng/ml, 20 ng/ml, 15 ng/ml, and 10 ng/ml.

Vitamin D deficiency defined using a cut off value of 30 ng/ml was associated with RDS (OR 3.478; 95% CI 1.817–6.659; $p < 0.001$; Fig 2A) in the random-effects model analysis. Among the included studies, there was no significant heterogeneity ($p = 0.517$; $I^2 = 0\%$). No single study affected the pooled result on sensitivity analysis (**S1-1 Fig in** S1 Fig) or cumulative analysis (**S1-2 Fig in** S1 Fig). Publication bias was not assessed since only two studies were included.

Vitamin D deficiency defined as a vitamin D level of less than 20 ng/ml was also associated with RDS (OR 4.549; 95% CI 3.007–6.881; $p < 0.001$; Fig 2B) in the random-effects model analysis. There was no significant heterogeneity ($p = 0.983$; I2 = 0%) among the included studies. No single study affected the result of the sensitivity analysis (**S2-1 Fig in** S2 Fig) or cumulative analysis (**S2-2 Fig in** S2 Fig). In the process of funnel plot inspection (**S2-3 Fig in** S2 Fig), asymmetry was detected, but there was no evidence of publication bias in either the Begg and Mazumdar rank-correlation test ($p = 0.452$) or Egger's regression test ($p = 0.131$).

Vitamin D deficiency defined as a vitamin D level of less than 15 ng/ml was associated with RDS (OR 17.267; 95% CI 1.084–275.112; $p = 0.044$; Fig 2C) in the random-effects model analysis. There was significant heterogeneity ($p = 0.006$; $I^2 = 86.8\%$) between the included studies (n = 2). We performed sensitivity analysis and cumulative analysis, but only two studies were included. We could not assess publication bias because of the small sample size.

Vitamin D deficiency defined as a vitamin D level of less than 10 ng/ml was also associated with RDS (OR 1.732; 95% CI 1.031–2.910; $p = 0.038$; Fig 2D) in the random-effects model analysis. There was heterogeneity ($p = 0.014$; $I^2 = 58.12\%$) among the included studies, and

(A) 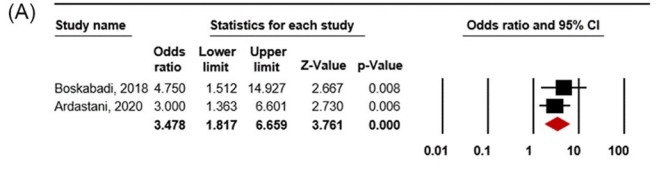

(B) 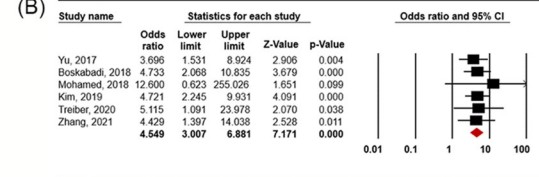

(C) 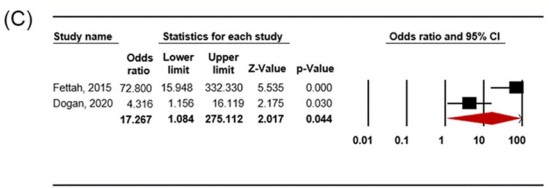

(D) 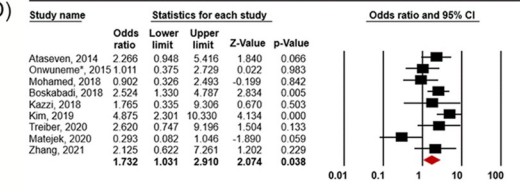

**Fig 2. Meta-analysis for the association between vitamin D deficiency and respiratory distress syndrome according to the definition of vitamin D deficiency.** (A) Vitamin D deficiency < 30 ng/ml, (B) Vitamin D deficiency < 20 ng/ml, (C) Vitamin D deficiency < 15 ng/ml, (D) Vitamin D deficiency < 10 ng/ml.

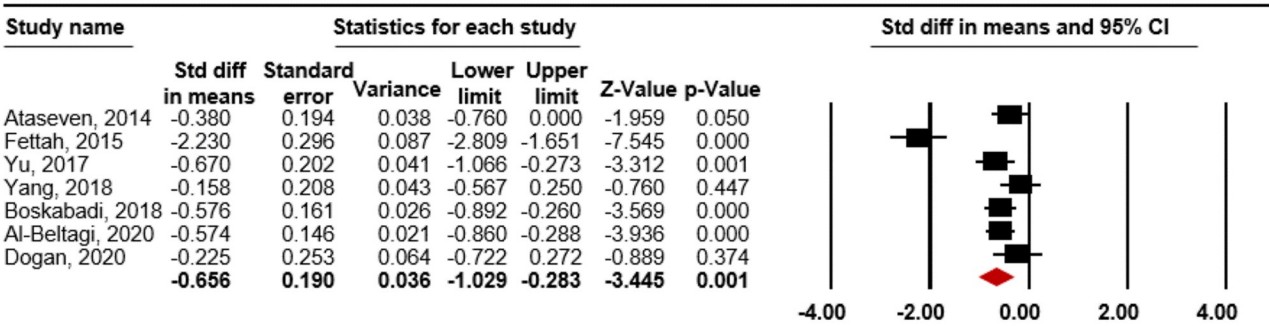

**Fig 3. Meta-analysis for the association between vitamin D level and respiratory distress syndrome.**

some studies affected the results of sensitivity analysis (**S3-1 Fig in** S3 Fig) and cumulative analysis (**S3-2 Fig in** S3 Fig). In the process of funnel-plot inspection, the presence of publication bias was not clear (**S3-3 Fig in** S3 Fig). In both the Begg and Mazumdar rank-correlation test ($p = 0.118$) and Egger's regression test ($p = 0.156$), there was no evidence of publication bias.

A lower level of vitamin D was also associated with RDS (SMD = -0.656; 95% CI -1.029 to -0.283; $p = 0.001$; Fig 3) in the random-effects model. On assessment of heterogeneity, there was significant heterogeneity among studies ($p < 0.001$; $I^2 = 83.50\%$), but the result of sensitivity analysis (**S4-1 Fig in** S4 Fig) or cumulative analysis (**S4-2 Fig in** S4 Fig) showed no significant change in the pooled results. On inspection of the funnel plot, we could not distinguish the presence of publication bias (**S4-3 Fig in** S4 Fig). Thus, we performed the Begg and Mazumdar rank-correlation test ($p = 0.368$) and Egger's regression test ($p = 0.202$), which showed no publication bias.

## Discussion

We found that vitamin D deficiency (defined using all existing criteria) was significantly associated with RDS development. Moreover, a lower level of vitamin D within 24 h of birth was associated with RDS. Thus, vitamin D deficiency may be a risk factor for RDS.

RDS reflects impaired or delayed surfactant synthesis and secretion, triggering pulmonary atelectasis and respiratory distress in preterm infants. The incidence of RDS decreases with increasing gestational age at birth, from 97% at 23 weeks of gestation to 65% at 28 weeks [33], 10.5% at 34 weeks, and 0.3% at 38 weeks [34]. The administration of antenatal corticosteroids to the mother and exogenous instillation of surfactant soon after birth are established preventative strategies for RDS [35]. The methods and timing of surfactant therapy have changed over the past decade. A thin catheter is used for prophylactic treatment of babies at high risk of RDS. Rescue therapy increasingly features early, nasal, continuous positive airway pressure and antenatal steroids [36].

In the clinical setting, vitamin D deficiency is frequently observed in pregnant women and infants [37–39]. Vitamin D deficiency in pregnant women is associated with preterm birth [40–42]. Moreover, vitamin D levels are lower in preterm infants than in term infants because less stored vitamin D is transferred transplacentally, and preterm infants have higher vitamin requirements [43]. Vitamin D levels at birth were lower in neonates born at less than 28 weeks of gestation [39], and vitamin D deficiency was more prevalent in neonates born at less than

32 weeks of gestation [38] than in more mature neonates. The vitamin D level of the fetus depends entirely on maternal circulation of vitamin D, and the level of 25(OH)D in the fetus is about two-thirds to three-quarters of the maternal level [44]. Levels measured within 24 h after birth (before supplementation), or in cord blood, correlate well with the levels in the mother and the fetus. The concentration of 25-(OH)D in blood is widely used as a biomarker reflecting vitamin D status because of the long half-life of 25-(OH)D [1, 45]. Most of the studies in this meta-analysis obtained vitamin D levels on the first day of life, but two studies did not describe the exact times of vitamin D measurements [26, 27].

No consensus definition of vitamin D deficiency has been established; studies have used different cutoff values when defining vitamin D deficiency [46]. The Institute of Medicine Committee and the American Academy of Pediatrics [47, 48] recommend that vitamin D levels should be maintained above 12 ng/ml (close to 20 ng/ml) for normal bone accretion in infants less than 1 year of age. In the study by Holick [1], a vitamin D level above 20 ng/ml was recommended.

Various hormones, including corticosteroids and thyroid hormones, are involved in surfactant synthesis [23]. Vitamin D, a steroid hormone, plays roles in surfactant synthesis and lung maturation [2, 3, 5, 11, 12]. Moreover, vitamin D deficiency itself has been associated with an increased need for assisted ventilation and a longer duration of ventilator support in preterm infants [24], and with development of BPD [13] as well as RDS. An adequate vitamin level may help to reduce the risk of RDS, BPD, and the need for ventilator support.

## Limitation of the study

There are several limitations to our study. First, each study used different diagnostic criteria when defining vitamin D deficiency, and small numbers of studies were thus included in the meta-analyses for the different cutoff values. This may have influenced the pooled effect. We were not able to assess the publication bias of studies with cutoff values of less than 30 ng/ml in terms of vitamin D deficiency. Second, various methods of measuring vitamin D levels in the blood have been used. Immunoassays such as enzyme-linked immunosorbent assays (ELISAs) and chemiluminescence assays, and liquid chromatography tandem mass spectrometry (LC-MS), were used to assess vitamin D status (Table 1). LC-MS is the gold standard method for 25-(OH)D measurement [43, 49]. Immunoassays yield higher vitamin D levels [43], and the performance thereof is not as good as that of LC-MS, especially at low vitamin D concentrations ($< 8$ ng/mL) [50]. The variabilities of vitamin D levels in this range seldom affect the identification of a vitamin D deficiency. Third, we could not control for any effect(s) of gestational age and/or the use of antenatal corticosteroids on RDS development; we could work only with the data from the included studies. Lastly, we found both heterogeneity and effects of individual studies on the results of sensitivity analysis and cumulative analysis for vitamin D deficiency $< 10$ ng/ml. Neonates included in the group with vitamin D $>10$ ng/ml could also have been affected by vitamin D deficiency ($< 30$ ng/ml or $< 20$ ng/ml). The use of different measuring methods for vitamin D could be why heterogeneity was present and the robustness of the sensitivity and cumulative analyses varied.

## Conclusion

This meta-analysis demonstrated an association between vitamin D deficiency or the vitamin D level within 24 h after birth, and the risk of RDS. In this meta-analysis, the included studies described the risk of RDS at various levels of vitamin D, thus vitamin D levels below 30 ng/ml, 20 ng/ml, 15 ng/ml, and 10 ng/ml respectively. Monitoring of neonatal vitamin D levels soon after birth and of maternal vitamin D levels during pregnancy, and maintaining adequate

vitamin D levels, may help to reduce the risk of respiratory morbidities including RDS and BPD [13] in preterm infants. Although, there is no consensus regarding what constitutes vitamin D deficiency or good vitamin D supplementation, vitamin D deficiency based on a level below 30 ng/ml showed an increased risk of RDS development in this meta-analysis. Thus, we cautiously assume that the optimal level of 25(OH)D should be higher than 30 ng/ml.

## Supporting information

**S1 Table. PRISMA checklist.**
(DOC)

**S2 Table. The quality assessment of included study using the Newcastle-Ottawa Scale.**
(DOCX)

**S1 Fig. Sensitivity analysis (1–1), and cumulative analysis (1–2) for the relationship between vitamin D deficiency defined using a cut off value of 30 ng/ml and RDS.**
(ZIP)

**S2 Fig. Sensitivity analysis (2–1), cumulative analysis (2–2), and funnel plot (2–3) for the relationship between vitamin D deficiency defined using a cut off value of 20 ng/ml and RDS.**
(ZIP)

**S3 Fig. Sensitivity analysis (3–1), cumulative analysis (3–2), and funnel plot (3–3) for the relationship between vitamin D deficiency defined using a cut off value of 10 ng/ml and RDS.**
(ZIP)

**S4 Fig. Sensitivity analysis (4–1), cumulative analysis (4–2), and funnel plot (4–3) for the relationship between vitamin D level and RDS.**
(ZIP)

## Author Contributions

**Conceptualization:** Yoo Jinie Kim, Hye Won Park.

**Data curation:** Yoo Jinie Kim, Gina Lim, Ran Lee, Hye Won Park.

**Investigation:** Gina Lim, Hye Won Park.

**Methodology:** Yoo Jinie Kim.

**Supervision:** Sochung Chung, Jae Sung Son.

**Validation:** Sochung Chung, Jae Sung Son.

**Writing – original draft:** Yoo Jinie Kim, Hye Won Park.

**Writing – review & editing:** Ran Lee, Sochung Chung, Jae Sung Son, Hye Won Park.

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
