## [Decision Letter · Decision Letter 0]

22 Sep 2022

PONE-D-21-40028Association between vitamin D level and respiratory distress syndrome: A systematic review and meta-analysisPLOS ONE

Dear Dr. Park,

Thank you for submitting your manuscript to PLOS ONE. After careful consideration, we feel that it has merit but does not fully meet PLOS ONE’s publication criteria as it currently stands. Therefore, we invite you to submit a revised version of the manuscript that addresses the points raised during the review process.

We look forward to receiving your revised manuscript.

Kind regards,

Roberta Hack Mendes, Ph.D.

Academic Editor

PLOS ONE

Journal Requirements:

Reviewers' comments:

Reviewer's Responses to Questions

**Comments to the Author**

1. Is the manuscript technically sound, and do the data support the conclusions?

Reviewer #1: Yes

Reviewer #2: Yes

2. Has the statistical analysis been performed appropriately and rigorously? 

Reviewer #1: Yes

Reviewer #2: Yes

3. Have the authors made all data underlying the findings in their manuscript fully available?

Reviewer #1: Yes

Reviewer #2: No

4. Is the manuscript presented in an intelligible fashion and written in standard English?

Reviewer #1: Yes

Reviewer #2: Yes

5. Review Comments to the Author

Reviewer #1: Thank you to the authors for this interesting study with results that can be applied to clinical practice. Overall, I couldn’t identify major changes that are needed. However I believe that two confusion biases might be addressed: gestational age and corticoid use during pregnancy. Please refer to the full review for details.

Reviewer #2: Abstract:

• The abstract is exemplary –a comprehensive but concise summary of background methods, results, and interpretation.

Introduction:

• The Introduction is very short – some material in the Discussion may be more useful in the Introduction to provide context for the study (see below).

• I think the Introduction does not strongly state what this meta-analysis can add to the existing body of knowledge and how it will be useful – it would be good to add a sentence on this.

Methods:

• “This meta-analysis was performed based on the reporting guidelines for systematic reviews and meta-analyses”: The PRISMA guidelines do not provide guidance on how to perform a review, only how it should be reported/written up. For this reason, I would recommend changing the phrasing to “This meta-analysis is reported in accordance with the reporting guidelines…” I would also name PRISMA specifically.

• The text states that two reviewers independently carried out screening, extraction, and quality assessment. For extraction, it is stated that discrepancies were resolved by discussion with a third reviewer. It would be good to clarify if this was the case for screening and quality assessment also.

• A fuller description of the Newcastle-Ottawa Scale would be helpful for readers who are unfamiliar with it, to clarify what each of the scales measure. A brief description of each scale would suffice.

• “We performed this meta-analysis to calculate a pooled estimate of how vitamin deficiency affected the occurrence of RDS”: The rest of the paper examines the association between VDD and RDS, without specifying that one causes the other or affects the other. I would change the phrasing in this sentence to refer to association, rather than one affecting the other, for consistency and to accurately describe the analysis.

Results:

• The findings are presented very clearly; the tables and figures are uniformly excellent and easy to read.

• First paragraph and Figure 1 refer to a “manual search”. What did the manual search involve? This should be explained in the Methods section. It would be good practice for a review to include forward and backward citation chasing (i.e. screening of papers that cite the included studies and screening of reference lists of included studies) – if this has been carried out, it’s an important part of the search strategy and should be mentioned.

• The meta-analysis appears to be appropriate. Stratifying studies by levels of VDD is appropriate, rather than combining all studies in a large meta-analysis. The difference in measurement methods is an important limitation but is duly noted in the limitations section.

• My only critique of the figures would be the labels for the forest plots. The meaning of the labels “Favours VDD” or “Favours RDS” with the + and – symbols isn’t very clear – these should be changed or explained in a note below each figure in a full sentence (e.g. “Favours VDD (+) = favours the existence of a positive association, such that greater VDD was associated with higher risk of RDS” or similar).

• It might be useful to include the full breakdown of scores (at an item level or scale level) for the quality assessment in the supplementary material. This provides greater transparency and justification for the scores presented in Table 1, and also allows the reader to quickly see what issues were examined in the quality assessment if they are not familiar with the tool.

Discussion:

• The information in the first three paragraphs is useful context and might be more useful in the introduction – presenting the existing evidence for a link between VDD and RDS and the basis for why this meta-analysis was carried out. The Discussion should primarily focus on the findings from this meta-analysis and comment only briefly how they fit into the existing research (as is done in the last sentence of paragraph 3 in the Discussion) – setting up the existing research should be done in the Introduction. Note that this is based on the conventions and journals I’m used to – if this journal has other standards for the different sections of the paper, feel free to disregard this comment.

• In paragraph 4, it is probably unnecessary to cite the statistics again – the details are in the Results section and the Discussion can focus on the broader picture and what those findings mean.

• The Discussion generally establishes the implications of the findings of the meta-analysis very well, without overstating. The section on limitations is thoughtful and state not only what the limitations were, but also what the impact of those limitations may have been on the findings.

General comments:

• The entire paper is extremely well-written; it is very concise but the methods are presented in comprehensive detail, the flow of ideas is logical, the results are presented with great clarity and are easy to follow, tables and figures are used extremely well, and the supplementary material is useful. I hope that my comments are helpful to refine the article and extend my best wishes to the authors.

• I will note that my expertise is in systematic reviews, not vitamin deficiency or respiratory medicine. For this reason my review focuses primarily on the methods and presentation of the findings. I have probably missed some nuances in the interpretation of the findings and how they stack up against existing work in the field, and perhaps some further limitations of the article that should be addressed by other peer reviewers with more specific knowledge of the area.

6. PLOS authors have the option to publish the peer review history of their article (what does this mean?). If published, this will include your full peer review and any attached files.

Reviewer #1: **Yes: **Taciane Alegra

Reviewer #2: No

---

## [Author Response · Author response to Decision Letter 0]

14 Oct 2022

Answer for editor’s comments: 

We have highlighted all of the revisions in yellow.

#1. Abstract

I believe it can be improved to synthesize the study better. Remember that most people will read the full text only if the abstract is interesting enough to call their attention. For example, some phrases from the introduction are better written than from the abstract and could be used to improve it. 

As you recommended, we have made the Abstract more interesting.

I also suggest adding that authors performed the meta-analysis model the reviewers used to synthesize the data, instead of citing which software they had used.

As you recommended, we have added text to the Abstract:

“For statistical analysis, we employed the random-effects model in Comprehensive Meta-Analysis Software ver. 3.3.”

In addition to that, I believe that the PRISMA 2020 checklist for abstracts could be used as a guide to improve the abstract.

We followed the PRISMA 2020 Abstract checklist. The details follow.

Title: Association between vitamin D level and respiratory distress syndrome: A systematic review and meta-analysis

Background (objectives): Growing evidence suggests an association between the vitamin D levels and respiratory outcomes of preterm infants. The objective of this systematic review and meta-analysis was to explore whether premature neonates with a vitamin D deficiency have an increased risk of respiratory distress syndrome (RDS).

Methods 

 Eligibility criteria: The search terms were ‘premature infant’, ‘vitamin D’, and ‘respiratory distress syndrome’. We retrieved randomized controlled trials and cohort and case-control studies.

 Information sources: We searched PubMed, EMBASE, and the Cochrane Library up through July 20, 2021.

Risk of bias: We employed the Newcastle-Ottawa Scales for quality assessment of the included studies.

Synthesis of results: For statistical analysis, we employed the random-effects model in Comprehensive Meta-Analysis Software ver. 3.3.

Results

 Included studies: A total of 121 potentially relevant studies were found, of which 15 (12 cohort studies and 3 case-control studies) met the inclusion criteria; the studies included 2,051 preterm infants. 

Synthesis of results: We found significant associations between RDS development in such infants and vitamin D deficiency within 24 h of birth based on various criteria, thus vitamin D levels < 30 ng/mL (OR 3.478; 95% CI 1.817–6.659; p < 0.001), < 20 ng/mL (OR 4.549; 95% CI 3.007–6.881; p < 0.001), < 15 ng/mL (OR 17.267; 95% CI 1.084–275.112; p = 0.044), and < 10 ng/ml (OR 1.732; 95% CI 1.031–2.910; p = 0.038), and an even lower level of vitamin D (SMD = –0.656; 95% CI –1.029 to –0.283; p = 0.001). 

Discussion 

 Limitation of evidence: Although the vitamin D deficiency definitions varied and different methods were used to measure vitamin D levels, vitamin D deficiency or lower levels of vitamin D within 24 h of birth were always associated with RDS development. 

Interpretation: Monitoring of neonatal vitamin D levels or the maintenance of adequate levels may reduce the risk of RDS.

#2. Introduction

The authors wrote “The level of vitamin D within 24 h of birth is associated with the risk of BPD in preterm infants [3]”, what gives an impression that this is well established in the literature, followed by: “Previous studies raised the possibility that the risk of respiratory distress syndrome (RDS) is increased in vitamin D-deficient preterm neonates, but the clinical relevance remains uncertain.” showing that this matter is controversial. Please, could the authors correct this disagreement? Also, could please authors provide citations to support the last quote?

As suggested by the Editor, we have corrected (and provide more details on) the “disagreement” issue.

#3. Results

1) Manual search - Could the authors please provide details about it? Does it mean that they searched the references?

We have added about it as follow; 

Additionally, we manually checked all reference lists in an effort to identify additional relevant studies.

2) In the flow chart (Fig 1) it is described that 2 articles were excluded due to “inadequate data for analysis” and 12 because “irrelevant data for analysis.” However, it was described in the methods that inclusion and exclusion criteria were based on design of the study, not on outcomes. So, please clarify that in the text.

We now write:

“Study design: A randomized controlled trial, or a cohort study (prospective or retrospective), or a case-control study; Patients and interventions (exposures): Newborns for whom vitamin D levels were known; Outcome: Respiratory distress syndrome (RDS).” 

Outcome: 

“The primary outcome was neonatal RDS diagnosed via chest radiographic findings and clinical presentations (e.g., tachypnea, nasal flaring, chest retraction, and cyanosis) within hours of birth.”

3) Regarding the 2 papers that had “inadequate data for analysis”, were they incomplete? Did the reviewers try to contact the study authors?

1) “25-OH Vitamin D deficiency in preterm babies: Incidence and association with morbidities” (Köksal et al.; poster below) 

This study did not compare the RDS incidence between infants with severe and moderate vitamin D deficiencies. The authors wrote: “There was no difference in RDS between severe and moderate vitamin D deficiency cases.” Thus, we excluded the poster.

2) “The potential effects of vitamin D deficiency on respiratory distress syndrome among preterm infants: (Al-Matary et al.) 

This paper included data on vitamin D deficiency, but the focus was on the surfactant doses for the vitamin D-deficient and normal groups, not the incidence of RDS. Thus, we excluded the paper. 

We did contact both authors by sending e-mails, but received no replies. 

4) Besides that, the primary outcome isn’t clear in the text, although one can infer that it is “respiratory distress syndrome”. It would be reasonable to state it clearly in the text, as the authors could study other outcomes linked to that (like death or need of surfactant treatment, for example).

We now describe the outcomes separately, as follows: 

 Outcomes

The primary outcome was neonatal RDS diagnosed via chest radiographic findings and clinical presentations (e.g., tachypnea, nasal flaring, chest retraction, and cyanosis) within hours of birth. 

#4. Characteristics of included studies

1) Regarding weight and gestational age, besides the mean, could authors please provide either standard deviation or the range for those data? They are relevant for clinicians reading the paper.

We have added the standard errors for gestational age and birth weight:

The mean birth weight of the study population in the 15 included studies was 1,833.2 g (standard error 143.91), and the mean gestational age of the infants at birth was 32.0 weeks (standard error 0.90); one study [28] did not describe gestational age at birth.

2)Table 1

- “Newcastle-Ottawa Scale for case-control study was used for study quality assessment, others used one for cohort study” - I couldn’t understand this phrase, could you please rephrase it?

We were seeking to state that different forms of the Newcastle-Ottawa scale were used (based on the type of study); thus, we have rephrased the sentence:

The two case-control studies were assessed using the Newcastle-Ottawa Scale for quality assessment of case-control studies; the other studies were assessed using the Newcastle-Ottawa Scale for the assessment of cohort studies

- In the line explaining the study of Kim et. al, could you please correct the format of the column with vitamin D deficiency definition? So it is in the same pattern as the others.

As recommended, we have changed the format of the column that lists the definitions of vitamin D deficiency in the line explaining the study of Kim et al. as follow; 

“severe: <10 ng/mL, deficiency: 10―20 ng/mL, insufficiency 20-30 ng/mL”

#5. Discussion

BPD is used in the text, but the authors didn’t make it clear that it refers to Bronchopulmonary dysplasia. Please add it on page 1, when the term first appears. 

As recommended, we now mention that BPD refers to bronchopulmonary dysplasia (first paragraph on page 1).

After reading the paper, two potential confusion biases still were not clear to me.

1. Influence of the gestational age - As the authors stated: “The incidence of RDS decreases with increasing gestational age at birth, from 97% at 23 weeks of gestation to 65% at 28 weeks [42], 10.5% at 34 weeks, and 0.3% at 38 weeks of gestation”. Thus, we know that gestational age itself can be a protective factor for developing RDS. Did the reviewers perform any subgroup analysis regarding gestational age? Would it be possible with the data available? Could vitamin D deficiency play different roles in different patient subsets?

We could not divide the studies by gestational age; thus, a subgroup analysis was not possible. This is a limitation of meta-analyses such as our study, and could be explored in future studies. We have added this as a limitation.

2. Corticoid administration during pregnancy -there is evidence showing that corticosteroid administration in preterm pregnancies can result in lower incidence of RDS. Did the studies included in the meta-analysis have controlled this factor? Did the reviewers consider that while doing the analysis? I believe both biases are relevant and should - at least - be cited in the discussion.

We also could not control for this factor because we could work only with the data from the included studies. We have added this as a limitation.

 

Answer for Reviewer #1 comments:

Thank you to the authors for this interesting study with results that can be applied to clinical practice. Overall, I couldn’t identify major changes that are needed. However, I believe that two confusion biases might be addressed: gestational age and corticoid use during pregnancy. Please refer to the full review for details.

We could not control for any effects of gestational age or antenatal steroid use on the development of RDS because we could work only with the data from the included studies. This is a limitation of meta-analyses. We have added this limitation. 

Answer for Reviewer #2 comments:

# Abstract: 

The abstract is exemplary –a comprehensive but concise summary of background methods, results, and interpretation.

We have revised the Abstract.

Growing evidence suggests an association between the vitamin D levels and respiratory outcomes of preterm infants. The objective of this systematic review and meta-analysis was to explore whether premature neonates with a vitamin D deficiency have an increased risk of respiratory distress syndrome (RDS). We searched PubMed, EMBASE, and the Cochrane Library up through July 20, 2021. The search terms were ‘premature infant’, ‘vitamin D’, and ‘respiratory distress syndrome’. We retrieved randomized controlled trials and cohort and case-control studies. 

For statistical analysis, we employed the random-effects model in Comprehensive Meta-Analysis Software ver. 3.3. We employed the Newcastle-Ottawa Scales for quality assessment of the included studies. A total of 121 potentially relevant studies were found, of which 15 (12 cohort studies and 3 case-control studies) met the inclusion criteria; the studies included 2,051 preterm infants. We found significant associations between RDS development in such infants and vitamin D deficiency within 24 h of birth based on various criteria, thus vitamin D levels < 30 ng/mL (OR 3.478; 95% CI 1.817–6.659; p < 0.001), < 20 ng/mL (OR 4.549; 95% CI 3.007–6.881; p < 0.001), < 15 ng/mL (OR 17.267; 95% CI 1.084–275.112; p = 0.044), and < 10 ng/ml (OR 1.732; 95% CI 1.031–2.910; p = 0.038), and an even lower level of vitamin D (SMD = –0.656; 95% CI –1.029 to –0.283; p = 0.001). Although the vitamin D deficiency definitions varied and different methods were used to measure vitamin D levels, vitamin D deficiency or lower levels of vitamin D within 24 h of birth were always associated with RDS development. Monitoring of neonatal vitamin D levels or the maintenance of adequate levels may reduce the risk of RDS.

# Introduction:

• The Introduction is very short – some material in the Discussion may be more useful in the Introduction to provide context for the study (see below).

• I think the Introduction does not strongly state what this meta-analysis can add to the existing body of knowledge and how it will be useful – it would be good to add a sentence on this.

We have revised and added the statement on introduction (below).

Vitamin D participates in mineral-ion homeostasis (e.g., calcium and phosphorus) and bone metabolism [1]. Apart from these classical functions, a role for vitamin D in lung development has been demonstrated by several studies (including animal works) [1-12]. Vitamin D regulates cell proliferation/differentiation, apoptosis, and angiogenesis in the lungs [6] and elsewhere in the body [1]. A recent meta-analysis found an association between the vitamin D level within 24 h of birth and the risk of bronchopulmonary dysplasia (BPD) in premature neonates [13]. 

Vitamin D receptors are expressed mainly in the lung during late pregnancy [2]; vitamin D thus affects lung functional and anatomical development, including cell differentiation and surfactant synthesis/secretion [3, 5, 14, 15]. Animal studies performed by Nguyen et al. [7-9] and Marin et al. [2, 16] using rat models revealed high-level expression of vitamin D receptor on type II alveolar cells at the end of gestation, when alveolar cell differentiation and surfactant synthesis commence [2, 16]. Rehan et al. [3] reported that the active form of vitamin D enhanced surfactant synthesis by type II pulmonary cells. Surfactant phospholipid and protein B synthesis increased in human pulmonary cancer cell lines. Such laboratory-based work raised the question of whether the respiratory distress syndrome (RDS) risk is increased in preterm neonates with vitamin D deficiencies in clinical settings. 

Several clinical studies have described low vitamin D levels in preterm infants who suffered from RDS [17, 18]. However, Matejek et al. [19] found no association between vitamin D deficiency and RDS. Given such conflicting evidence, we performed a systematic review and meta-analysis to explore whether vitamin D deficiency or the vitamin D level measured within 24 h of birth was associated with an increased risk of RDS..

# Methods:

• “This meta-analysis was performed based on the reporting guidelines for systematic reviews and meta-analyses”: The PRISMA guidelines do not provide guidance on how to perform a review, only how it should be reported/written up. For this reason, I would recommend changing the phrasing to “This meta-analysis is reported in accordance with the reporting guidelines…” I would also name PRISMA specifically.

We have revised the text: 

“This meta-analysis was reported based the Preferred Reporting Items for Systematic Reviews and Meta-Analyses (PRISMA) recommendations”

• The text states that two reviewers independently carried out screening, extraction, and quality assessment. For extraction, it is stated that discrepancies were resolved by discussion with a third reviewer. It would be good to clarify if this was the case for screening and quality assessment also.

We have revised the description of screening, data extraction, and quality assessment as follows:

In the “Search strategy and study selection” section: 

“We initially reviewed the titles and abstracts of the articles, and then reviewed the full-text articles. The reviews were performed independently by two reviewers (HW Park and Y Kim) using criteria that we determined before the review process for inclusion in the meta-analysis. Any disagreement during the process was resolved by the third author (R Lee).” 

In the “Data extraction” section:

“If there were any discrepancies during the review process with two authors (YJ Kim and HW Park), we discussed these with the third reviewer (R Lee) and we reviewed the study again.” 

In the “Study quality assessment” section: 

“We (Y Kim and HW Park) also separately assessed the quality of included studies using the Newcastle–Ottawa Scale (NOS) [22]. Disparities in the assessment were resolved through discussion with the third author (R Lee).” 

• A fuller description of the Newcastle-Ottawa Scale would be helpful for readers who are unfamiliar with it, to clarify what each of the scales measure. A brief description of each scale would suffice.

We have added the items included in the Newcastle-Ottawa Scales and we now describe the quality assessment results in detail in the Supplementary Material (S2 Table) 

• “We performed this meta-analysis to calculate a pooled estimate of how vitamin deficiency affected the occurrence of RDS”: The rest of the paper examines the association between VDD and RDS, without specifying that one causes the other or affects the other. I would change the phrasing in this sentence to refer to association, rather than one affecting the other, for consistency and to accurately describe the analysis.

We now mention the manual search in the Methods: 

 “We performed this meta-analysis to calculate a pooled estimate of odds ratio (OR) for the association between vitamin D deficiency and the occurrence of RDS.”

# Results:

• The findings are presented very clearly; the tables and figures are uniformly excellent and easy to read.

• First paragraph and Figure 1 refer to a “manual search”. What did the manual search involve? This should be explained in the Methods section. It would be good practice for a review to include forward and backward citation chasing (i.e. screening of papers that cite the included studies and screening of reference lists of included studies) – if this has been carried out, it’s an important part of the search strategy and should be mentioned.

We have explained about manual search in Method section as follow; 

Additionally, we manually checked reference lists to identify relevant studies for this meta-analysis.

• The meta-analysis appears to be appropriate. Stratifying studies by levels of VDD is appropriate, rather than combining all studies in a large meta-analysis. The difference in measurement methods is an important limitation but is duly noted in the limitations section.

We have revised the existing text and added new text as follows: 

Second, various methods of measuring vitamin D levels in the blood have been used. Immunoassays such as enzyme-linked immunosorbent assays (ELISAs) and chemiluminescence assays, and liquid chromatography tandem mass spectrometry (LC-MS), were used to assess vitamin D status (Table 1). LC-MS is the gold standard method for 25-(OH)D measurement [45, 51]. Immunoassays yield higher vitamin D levels [45], and the performance thereof is not as good as that of LC-MS, especially at low vitamin D concentrations (< 8 ng/mL) [52]. The variabilities of vitamin D levels in this range seldom affect the identification of a vitamin D deficiency.

• My only critique of the figures would be the labels for the forest plots. The meaning of the labels “Favours VDD” or “Favours RDS” with the + and – symbols isn’t very clear – these should be changed or explained in a note below each figure in a full sentence (e.g. “Favours VDD (+) = favours the existence of a positive association, such that greater VDD was associated with higher risk of RDS” or similar).

We have removed the figure labels. A positive OR means that vitamin D deficiency was associated with a risk of RDS. 

• It might be useful to include the full breakdown of scores (at an item level or scale level) for the quality assessment in the supplementary material. This provides greater transparency and justification for the scores presented in Table 1, and also allows the reader to quickly see what issues were examined in the quality assessment if they are not familiar with the tool.

We have added the items included in the Newcastle-Ottawa Scales and we now describe the detailed results of our quality assessment in the S2 Table (in addition to the total scores in Table 1).

# Discussion:

• The information in the first three paragraphs is useful context and might be more useful in the introduction – presenting the existing evidence for a link between VDD and RDS and the basis for why this meta-analysis was carried out. The Discussion should primarily focus on the findings from this meta-analysis and comment only briefly how they fit into the existing research (as is done in the last sentence of paragraph 3 in the Discussion) – setting up the existing research should be done in the Introduction. Note that this is based on the conventions and journals I’m used to – if this journal has other standards for the different sections of the paper, feel free to disregard this comment.

We have changed both the Introduction and Discussion. Some findings mentioned in the Discussion are now addressed in the Introduction to support the aim of the study.

• In paragraph 4, it is probably unnecessary to cite the statistics again – the details are in the Results section and the Discussion can focus on the broader picture and what those findings mean.

We have removed the statistical details from the Discussion. 

The revision reads:

“We found that vitamin D deficiency (defined using all existing criteria) was significantly associated with RDS development. Moreover, a lower level of vitamin D within 24 h of birth was associated with RDS. Thus, vitamin D deficiency may be a risk factor for RDS.”

• The Discussion generally establishes the implications of the findings of the meta-analysis very well, without overstating. The section on limitations is thoughtful and state not only what the limitations were, but also what the impact of those limitations may have been on the findings.

We appreciate the kind comments that we received.

---

## [Decision Letter · Decision Letter 1]

1 Dec 2022

Association between vitamin D level and respiratory distress syndrome: A systematic review and meta-analysis

PONE-D-21-40028R1

Dear Dr. Park,

We’re pleased to inform you that your manuscript has been judged scientifically suitable for publication and will be formally accepted for publication once it meets all outstanding technical requirements.

Kind regards,

Roberta Hack Mendes, Ph.D.

Academic Editor

PLOS ONE

Additional Editor Comments (optional):

Reviewers' comments:

Reviewer's Responses to Questions

**Comments to the Author**

1. If the authors have adequately addressed your comments raised in a previous round of review and you feel that this manuscript is now acceptable for publication, you may indicate that here to bypass the “Comments to the Author” section, enter your conflict of interest statement in the “Confidential to Editor” section, and submit your "Accept" recommendation.

Reviewer #1: All comments have been addressed

2. Is the manuscript technically sound, and do the data support the conclusions?

Reviewer #1: Yes

3. Has the statistical analysis been performed appropriately and rigorously? 

Reviewer #1: Yes

4. Have the authors made all data underlying the findings in their manuscript fully available?

Reviewer #1: Yes

5. Is the manuscript presented in an intelligible fashion and written in standard English?

Reviewer #1: Yes

6. Review Comments to the Author

Reviewer #1: Thank you very much to the authors for accepting the suggestions and making some changes in the text. This revision is well written and pleasant to read, my compliments to the authors.

7. PLOS authors have the option to publish the peer review history of their article (what does this mean?). If published, this will include your full peer review and any attached files.

Reviewer #1: **Yes: **Taciane Alegra

---

## [Editor Report · Acceptance letter]

4 Dec 2022

PONE-D-21-40028R1 

Association between vitamin D level and respiratory distress syndrome: A systematic review and meta-analysis 

Dear Dr. Park:

I'm pleased to inform you that your manuscript has been deemed suitable for publication in PLOS ONE. Congratulations! Your manuscript is now with our production department. 

Kind regards, 

on behalf of

Dr. Roberta Hack Mendes 

Academic Editor

PLOS ONE